# Configurable Pseudo Noise Radar Imaging System Enabling Synchronous MIMO Channel Extension

**DOI:** 10.3390/s23052454

**Published:** 2023-02-23

**Authors:** Niklas Bräunlich, Christoph W. Wagner, Jürgen Sachs, Giovanni Del Galdo

**Affiliations:** 1Electronic Measurements and Signal Processing Group, Technische Universität Ilmenau, 98693 Ilmenau, Germany; 2Fraunhofer Institute for Integrated Circuits IIS, 98693 Ilmenau, Germany; 3Ilmsens GmbH, 98693 Ilmenau, Germany

**Keywords:** ultra-wideband, PN radar, synchronous MIMO, system architecture, multichannel sensing, Red Pitaya

## Abstract

In this article, we propose an evolved system design approach to ultra-wideband (UWB) radar based on pseudo-random noise (PRN) sequences, the key features of which are its user-adaptability to meet the demands provided by desired microwave imaging applications and its multichannel scalability. In light of providing a fully synchronized multichannel radar imaging system for short-range imaging as mine detection, non-destructive testing (NDT) or medical imaging, the advanced system architecture is presented with a special focus put on the implemented synchronization mechanism and clocking scheme. The core of the targeted adaptivity is provided by means of hardware, such as variable clock generators and dividers as well as programmable PRN generators. In addition to adaptive hardware, the customization of signal processing is feasible within an extensive open-source framework using the Red Pitaya^®^ data acquisition platform. A system benchmark in terms of signal-to-noise ratio (SNR), jitter, and synchronization stability is conducted to determine the achievable performance of the prototype system put into practice. Furthermore, an outlook on the planned future development and performance improvement is provided.

## 1. Introduction

Emerging from the twentieth century, a broad variety of surface-penetrating radar imaging technologies have been developed, the fundamental objective of which is to examine optically opaque scenarios under test (SUTs) by means of microwaves. Within a frequency range of up to a few GHz, the penetration of electromagnetic waves into different types of matter often works quite well, so that microwave imaging techniques can be utilized for several purposes, such as the detection of hidden or buried objects, visualization of material inhomogeneity, or investigation of concealed processes. That is why its application can be seen in many fields, such as geosciences, industrial quality assurance processes, non-destructive testing (NDT) and in medical engineering.

Microwave imaging being applied in the context of feature extraction from stacked layers of soil, just as in the context of spatially localizing buried objects, is referred to as ground-penetrating radar (GPR). The GPR principle is applied in areas such as archaeology [1] and agriculture [2]. In addition, GPR sensors are also used in the retrieval of buried victims within avalanches [3] or earthquake debris [4]. Conventional GPR devices are typically based on one pair of transmitter (Tx) and receiver (Rx) antennas, which is scanned over the area of interest. In order to reduce the scanning effort due to the necessary meandering, one also finds system extensions, where multiple antenna pairs are stacked to a linear array [5].

However, not only for GPR applications, the spatial detection of objects, movements, or inhomogeneities is a desired functionality. Additionally, in medical imaging, the radar principle can be used to screen through layers of human tissue. That way, using conformal antenna arrays, arterial movement covered by the human skull may become visible [6], or cancerous tissue may be found by employing magnetically stimulated nano particles [7,8].

In this work, we refer to an arbitrarily arranged antenna array—planar or conformal—with multiple transmit and receive elements in unipolar or full-polarimetric orientation—Figure 1 symbolizes two possible applications. Additionally, we limit ourselves to a measurement volume of interest in close proximity of the antennas, as it is usual in subsurface-penetrating radar. The array is working in multiple-input and multiple-output (MIMO) mode. That is, every receiving antenna is able to capture the backscattered waves launched from every individual transmitter. In addition to the aforementioned reduction in the scanning effort, such arrays allow for better target classification, since the waves scattered from every targets are registered for different angles of incidence and reflection as well as for different polarizations. Furthermore, a fast-operating full antenna array enables spatial and temporal tracking of time-variant targets.

### 1.1. System Model

For the remainder of this article, the symbols w, W, W, ∗, and ⊛ denote vectors, matrices, tensors, convolution, and circular (periodic) convolution, respectively. It is known from linear system theory that any signal ξ(t) can be represented by a series of values $\xi \left[n\right]=\xi \left(n\;\cdot \;T_{0}\right)$ at a uniform sampling rate of f0=1T0, as long as the Nyquist–Shannon sampling theorem is fulfilled. In addition, the following signal discretizations apply:$$\begin{array}{cccccc} x(t) & \;↔\; & x[n] & \;\mathrm{or}\; & x & \in C^{N},\\ y(t) & \;↔\; & y[n] & \;\mathrm{or}\; & y & \in C^{N},\\ h(\tau ) & \;↔\; & h[\nu ] & \;\mathrm{or}\; & h & \in C^{N}\;\mathrm{and}\\ \delta (t) & \;↔\; & \delta [n] & \;\mathrm{or}\; & \delta & \in {\left\{0,1\right\}}^{N}, \end{array}$$
where δ is the Kronecker delta. For the sake of simple notation, we further define that periodic signals ξ(t) of period $T_{N}=N\;\cdot \;T_{0}$ can be represented by a vector $\xi \in C^{N}$, holding the discretized signal over one period.

Additionally, in the further course of this article, the term ultra-wideband (UWB) classifies systems or signals of high fractional bandwidth of more than 20% of the carrier frequency as a result of high absolute bandwitdth in baseband, ranging from a few hundred MHz to several GHz[10].

Assuming a radar array composed of a number of *K* Tx and a number of *L* Rx antennas, the measurement result of all the antennas transmitting and receiving in parallel may be summarized by the following equation: (1)$$\begin{array}{ccc} \left[\begin{array}{c} y_{1}\\ y_{2}\\ \vdots \\ y_{L} \end{array}\right] & =\left[\begin{array}{cccc} h_{11} & h_{12} & \ldots & h_{1K}\\ h_{21} & h_{22} & \ldots & h_{2K}\\ \vdots & \vdots & \ddots & \vdots \\ h_{L1} & h_{L2} & \ldots & h_{LK} \end{array}\right]\; & ⊛\left[\begin{array}{c} x_{1}\\ x_{2}\\ \vdots \\ x_{K} \end{array}\right]\\ Y & =H⊛X, \end{array}$$
where $Y\in C^{L\times N}$ denotes the set of *L* receive signals, $X\in C^{K\times N}$ denotes the set of *K* transmit signals, and $H\in C^{L\times K\times N}$ denotes the full MIMO model of the L×K impulse responses $h_{\ell k}$, representing the path between all the possible pairs of receive and transmit antennas [10]. The (circular) correlation occurs along the axis corresponding to the periodic time domain signal with period *N*.

In order to be able to determine the full measurement tensor H of all the impulse responses in parallel, the stimulus signals $x_{k}$ emitted by each of the transmit antennas must be uncorrelated, i.e., $x_{k}^{H}⊛x_{k^{\prime }}=\delta \;\forall k,k^{\prime }\in 1,\cdots ,K$. If uncorrelated stimulus signals are not available, the measurements have to be repeated *K* times, whereat the Tx antennas are stimulated sequentially, one at a time, resulting in a slight modification of (Equation 1): (2)$$\begin{array}{cc} \left[\begin{array}{cccc} y_{11} & y_{12} & \ldots & y_{1K}\\ y_{21} & y_{22} & \ldots & y_{2K}\\ \vdots & \vdots & \ddots & \vdots \\ y_{L1} & y_{L2} & \ldots & y_{LK} \end{array}\right] & =\left[\begin{array}{cccc} h_{11} & h_{12} & \ldots & h_{1K}\\ h_{21} & h_{22} & \ldots & h_{2K}\\ \vdots & \vdots & \ddots & \vdots \\ h_{L1} & h_{L2} & \ldots & h_{LK} \end{array}\right]⊛\left[\begin{array}{cccc} x_{1} & 0_{N} & \ldots & 0_{N}\\ 0_{N} & x_{2} & \ldots & 0_{N}\\ \vdots & \vdots & \ddots & \vdots \\ 0_{N} & 0_{N} & \ldots & x_{K} \end{array}\right]\\ Y^{\prime } & =H⊛X^{\prime }, \end{array}$$
where $Y^{\prime }\in C^{L\times K\times N}$, $X^{\prime }\in C^{K\times K\times N}$, and $0_{N,\cdots }\in {\left\{0\right\}}^{N,\cdots }$. Evidently, the measurement process behind (Equation 2) requires *K* times more measurements to retrieve H; however, at the same time, the cross-correlation requirement of (Equation 1) can be relaxed to $x_{k}^{H}⊛x_{k}=\delta \;\forall k\in 1,\cdots ,K$, which is of high practical relevance.

Many imaging applications (including radar) share the common goal of retrieving hidden information about an object’s internal geometric structure from the surface observations, typically collected in the form of MIMO system impulse responses, such as H. In applying the Born approximation [11], where the volume of interest may only contain a limited set of ρ point scatterers while also excluding multipath reflections and direct antenna-to-antenna-coupling, $h_{\ell k}\left(\tau \right)$ may be approximated as: (3)$$h_{\ell k}\left(\tau \right)=\sum_{\rho =1}^{P}\frac{A_{k}\left(\tau \right)*\delta \left(\tau -\frac{r_{k\rho }}{c}\right)}{2\pi r_{k\rho }}*\Lambda_{\rho }\left(\tau \right)*\frac{B_{\ell }\left(\tau \right)*\delta \left(\tau -\frac{r_{\ell \rho }}{c}\right)}{2\pi r_{\ell \rho }},$$
where τ denotes the time parameter for impulse response functions (IRFs) and *c* corresponds to the medium’s propagation speed. For each of the *P* scattering objects, $r_{\ell \rho k}$ denotes the total round-trip distance from the transmit antenna *ℓ*, target, and receive antenna *k*. Finally, $A_{k}\left(\tau \right)$, $B_{\ell }(\tau$), and $\Lambda_{\rho }\left(\tau \right)$ denote the IRF of the transmit antenna, receive antenna, and scattering target ρ, respectively. Note that the notation of (Equation 3) does not the respect angular dependencies of antenna radiation and scattering as well as the polarimetric behavior (see [10] for details).

As recognizable from (Equation 3), an important step of the imaging approach is to compensate for the different antenna–target–propagation delays, which is elaborated in detail, e.g., in [12,13]. Though not going into detail here, this motivates one crucial aspect of the MIMO imaging approach, which is the importance of a correct synchronization between the measurement channels of the array.

Furthermore, we assume that the sut comprises two types of targets: targets with (observation) time-invariant scattering (e.g., scattering from IEDs buried in shallow depths [14]) as $\Lambda_{\alpha }\left(\tau \right)$, and targets with (observation) time-variant scattering as $\Lambda_{\beta }\left(\tau ,t^{\prime }\right)$, where $\;\cdot \;(t^{\prime })$ is the time dimension along which the model H changes (typically referred to as *slow time*), and is under no circumstances to be confused with the ·(t) time domain (also referred to as *fast time*) that is introduced earlier alongside signal discretizations. Examples of time-variant scenarios include living (i.e., breathing) persons, covered by avalanches or hidden behind walls [15,16], or the detection of wood-decomposing parasites, such as termites or bark beetles [6].

Ultimately, this distinction leads to the separation of
(4)$$H\left(t\right)=H_{\alpha }+H_{\beta }\left(t\right),$$
where $H_{\alpha }$ and $H_{\beta }\left(t\right)$ denote the time-invariant and time-variant components, respectively. Please note that the argument further discretizing H(t) adds another time axis. Oftentimes, only the time-invariant components of H are of interest while the time-variant components hold unwanted clutter of spurious sources. If the time-variant components $H_{\beta }$ can reasonably be assumed as the result of an unbiased random process, $H_{\alpha }$ can be approximated from averaging *R* measurements of H: (5)$$H_{\alpha }\approx \frac{1}{R}\left(\sum_{r=1}^{R}H\left(t_{R}\right)\right),\;\mathrm{assuming}\;\;\lim_{R\to \infty }∥\frac{1}{R}\left(\sum_{r=1}^{R}H_{\beta }\left(t_{R}\right)\right)∥=0,$$
where $t_{R}$ denotes the time instance of each individual measurement of H(t). This operation is crucial with respect to random timing errors (jitter) of the measurement electronics, especially if the mostly weakly scattering time-variant targets are located close to a strong time-invariant object. While random jitter hardly affects signal parts with constant amplitude, it causes severe noise at steep signal edges. Therefore, a strong target in connection with a jittering sensor device may causes a local elevation of noise, which overwhelms the scattering from a weak moving target [6].

### 1.2. The Importance of Synchronization

In addition, to correct microwave imaging, the emphasis of time-variant targets is another important reason for a precise synchronization of the MIMO antenna array. That is, for both of the above-mentioned cases (Equation 1) and (Equation 2), each transmitting of a stimulus signal $x_{k}\left(t\right)$ as well as each reception of a system response $y_{\ell }\left(t\right)$ must align precisely upon a common time base for the system to operate in unison and for the resulting data to be coherent. Therefore, there are essentially two approaches to the synchronization of MIMO measurements:*Retrospective synchronization*: The measurements are carried out asynchronously (sequentially), whereby, e.g., different time bases or time domain signal offsets occur between different channels, which must be subsequently compensated for, to achieve coherent data analysis.*A priori synchronization*: It is ensured in advance that all transmitters and receivers are synchronized to the same time base and that the measurement is started synchronously at all channels as well as all that channels have the same displacement in time domain.

Examples for the application of retrospective synchronization in the context of UWB radar can be found in  [5,7]. Some approaches of retrospective synchronization use pivot signals, where a once-generated reference signal is distributed and fed into each receiver signal path $y_{\ell }\left(t\right)$. Once its relation to the locally unsynchronized distributed measurement node is known, it is possible to compensate for its effect. This principle is shown in [17] for an M-sequence radar application and in [18] for UWB angular sensing using universal software receivers.

Our approach to UWB radar implements a priori synchronization in combination with an evolved distributed multichannel M-sequence system architecture, which is capable of fully configuring each measurement node in terms of transmit signals and sampling clocks. This results in a highly flexible system capable of coherent multichannel measurements, thus challenging the known preconception in the imaging community that *there is not the one universal sensor* by handing more (synchronous) control to the user. Further, due to the full coherence, our concept can be utilized in implementing alternative sampling strategies, such as Compressive Sensing (CS) UWB radar [19].

All (distributed) signal transmitters and receivers are aligned precisely to within one main clock interval $T_{0}$, which is only a few picoseconds long. Synchronization can be triggered from a single control signal, which may originates from signal sources with high timing uncertainties, such as embedded processors, by means of a “helper” clock running at a sufficiently low rate to capture the control system, but of similar timing performance as the main clock. Thus, the edges of the resulting synchronization signal can be distributed to synchronize each individual measurement node in the system.

The remainder of this article is organized as follows. In Section 2, the reader is familiarized to the state-of-the-art UWB pseudo-random noise (PRN) sensor architecture and the key aspects that have an impact on achievable system performance are discussed. In addition, it is elaborated which requirements have to be satisfied for these sensors to be capable of the MIMO mode of operation. Therefore, as an essential part of the targeted true MIMO synchronization, a novel UWB PRN generator integrated circuit (IC) is presented. Thereafter, in Section 3, the fundamental array node of the proposed design is presented and its most significant features are stressed. Special attention is paid to the underlying synchronization mechanism, which, by the help of the introduced programmable PRN generators and clock dividers, enables true MIMO scalability of the system. In Section 4, characterization measurements are performed on signal-to-noise ratio (SNR), jitter, and synchronization stability. From these, predictions can be provided about the obtainable system performance. Finally, Section 5 concludes this article.

## 2. PRN UWB Sensors for MIMO Imaging

While there are essentially two well-established UWB sensor concepts that mainly vary in excitation signals (pulse excitation and PRN stimulation), this article focuses on PRN concepts. Due to the large fractional bandwidth of the sounding signal and the rigid synchronization of Tx and Rx channels, UWB PRN sensors are well suited for MIMO imaging. Here, we assume a fractional bandwidth larger than 100%.

### 2.1. M-Sequence UWB Devices

Binary PRN provides good prerequisites to meet both conditions—bandwidth and synchronicity—based on a simple system concept. There are several types of PRN signals, each with specific characteristics. For the remainder of this article, we primarily choose the excitation signals $x_{k}$ to be maximum-length binary sequences (MLBSs) (also called M-sequences), due to their tremendous advantage of exhibiting an autocorrelation spectrum approximating a Dirac function
(6)$$C_{xx,k}=x_{k}⊛x_{k}\approx \delta .$$

Moreover, of the various UWB radar techniques, the M-sequence radar is the optimum when considering the bandwidth, measurement speed, implementation cost, or MIMO enhancement in total [10]. M-sequences are wideband, periodically recurring, binary sequences that appear random at first glance but are, in fact, strictly deterministic. They thus combine the properties of periodic–deterministic and random signals.

The state-of-the-art M-sequence sensor approach as of [5,6,20,21,22], which is, however, not yet fully synchronizable (i.e., it lacks controllable reset mimics) in terms of MIMO applications, is presented in Figure 2 and some signal aspects are provided in Figure 3.

Driven by a stable radio frequency (RF) clock at rate $f_{0}$, the output sequences $x_{k}$ are generated in a so-called linear feedback shift register (LFSR). Depending on the requirements of the intended application, typical values of $f_{0}$ are to be found in the approximate range of 1–50 GHz. Per period *T*, the LFSR of length *m* produces $N=2^{m}-1$ chips. Since plain sampling according to the Nyquist criterion would require enormous sampling rates, interleaved subsampling may be employed in data acquisition as a reasonable trade-off between the measurement time and analog-to-digital converter (ADC) specifications. That is—assuming $x_{k}\left(t\right)$ being band-limited to $f_{0}\;/\;2$—one data sample per chip is sufficient to meet the Nyquist sampling criterion or the equivalent sampling rate can be equal to the clock rate $f_{eq}=f_{0}$. Hence, the actual sampling clock $f_{S}$ can be obtained by a divide-by-*S* clock divider controlling the timing relations between Tx and Rx circuitry [10].

The system response $y_{\ell }\left(t\right)$ is captured by a wideband track and hold (T&H) device and subsequently digitized by the ADC operating at $f_{S}$. For convenience, their clock signals are of identical origin in Figure 2. However, as presented later in this article, this may be implemented in alternative ways as well. Basically, the range of $f_{S}$ can be selected arbitrarily, as long as *S* does not divide the period of the transmit signal. This constraint is necessary to enforce the correct equivalent sampling. However, the larger $f_{S}$ is, the faster the measurement will be. The rate to capture one IRF is typically determined by the time variance (i.e., scanning speed or variation rate of time-variant targets) of the sut. If $f_{S}$ is selected as sufficiently large, the actual measurement rate may be higher than required. At this point, it is therefore quite useful to cache and pre-process the digitized raw data $b_{\ell }$ at a pre-processor in close vicinity to the recording antenna. This way, by applying one of several data reduction techniques, the resulting amount of data ${\tilde{b}}_{\ell }$ can be significantly decreased. In a state-of-the-art MLBS radar, synchronous averaging (“stacking”) is primarily used for data reduction as it has the positive side effect of reducing the noise floor by a factor $\sqrt{p}$, where *p* is the number of averages.

Even though the captured data are pre-processed in terms of data reduction, the resulting signal ${\tilde{b}}_{\ell }$ is still difficult to interpret, since it is spread over the whole (propagation) time interval. It has, therefore, to be compressed before it can be considered as a typical radar signal. From (Equation 6), it follows that the impulse response function ${\tilde{h}}_{\ell k}$ can be estimated by the cross-correlation of the digitized (and pre-processed) receive signal ${\tilde{b}}_{\ell }$ with the ideal discretized transmit signal $x_{k}$ as
(7)$${\tilde{h}}_{\ell k}\approx {\tilde{b}}_{\ell }⊛x_{k}.$$

Practically, this can be implemented numerically, e.g., using the fast Hadamard transform
(FHT), which is a fast algorithm that is organized similarly to the fast Fourier transform
(FFT) but only needs addition operations [23].

Based on its mode of operation, the signals in the radar circuitry shown in Figure 2 may be divided into two levels of timing speed, precision, or synchronicity:*Picosecond precision*: The edges of the LFSR clock signal as well as the edges of the T&H clock should be as steep and as stable as possible, since they determine the random sampling jitter independently of their clock rate being in the GHzrange (LFSR) or the 10 MHz range (T&H).*Nanosecond precision*: Assuming a stable hold phase of the T&H circuit, the timing precision of the ADC clock and the digital pre-processing is less critical.

Apart from these signal precision levels, the output of the pre-processing covers a reduced amount of data, which can be cached before asynchronous transmission to a host unit.

As also recognizable from Figure 2, the M-sequence concept provides a very rigid synchronization since, first, both the transmitter and receiver are connected to the same stable clock source, and, second, all timing-critical signals exhibit steep edges (about 20.00 ps rise time), which minimizes the generation of sampling and PRN jitter. Furthermore, the impulse compression (Equation 7) spreads the residual jitter equally over the signal avoiding any noise accumulation at steep edges of ${\tilde{h}}_{\ell k}$ [6].

### 2.2. Multichannel UWB Sensor Design

Considering the state-of-the-art MLBS sensor that is introduced in Figure 2, all its various system properties and blocks may be abstracted into specific functions to develop a more general architecture. In this way, the system architecture can be made scalable so that the radar array is built from a number *Q* of electrically and mechanically identical basic units, called the *“measurement node”*, to achieve maximum user flexibility. As shown in Figure 4, every measurement node contains two Tx and two Rx channels, so that it may be operated as a full-polarimetric array element. These elements may be arbitrarily geometrically placed within the sensor array. Note that there are no restrictions in the case of UWB arrays—fractional bandwidth greater than 100% supposed—with respect to the maximum element spacing for grating loop suppression as the λ2-limit for narrowband arrangements. Based on Figure 4, a measurement node can be understood as a subsystem and the main functional properties of which are:The transmitter and receiver unit that accomplishes the analog RF signal tasks that are most likely the generation and conditioning of UWB PRN stimulus signals as well as the receiving of the sut’s system responses using T&H circuits.The digitizing and pre-processing unit that performs analog-to-digital conversion and hardware–level signal processing as well as the streaming of recorded data to a secondary processing or storage unit.

In total, the sensor array is composed of K=L=2Q transmitter and receiver channels, so that the array is able to capture $4Q^{2}$ impulse responses within one measurement frame in maximum.

However, in order that each measurement node operates with precise and rigid timing, a certain number of clock and control signals are necessary. These are essentially the fast M_CLK, driving the shift registers for output sequence generation at a rate of $f_{0}$, and the divided clock DIV_CLK, controlling the sampling at a rate of $f_{S}$. In addition, a SYNC signal becomes mandatory to synchronize all units and trigger acquisition at each measurement node, respectively. Hence, to conceptualize a multinode imaging system, one must add a clocking and synchronization unit that provides these signals. Concluding thereupon, the minimum measurement configuration (designated as *“standalone measurement node”*) is built from at least *one* sensor node, a secondary processing or data host unit, and the composite of clock, synchronization, and control units.

As the array is able to operate in different modes, in which all receiver channels are always active and operating in parallel but the transmitter channels may be switched off in certain constellations, the bus interface to each Tx chip must be provided by the control unit. This way, each Tx channel may be changed in terms of PRN stimulus or its sequence outputs are totally silenced. Concerning the fully equipped radar array built from an arbitrary number of measurement nodes, the following operation modes are foreseen:*Sequential mode*: In this mode, all transmitters are operated with an identical PRN signal, which is typically an M-sequence due to its best autocorrelation properties. Since there are several M-sequences per order, which are, however, strongly correlated with each other, the different transmitters must be operated sequentially according to (Equation 2), while all receivers always operate in parallel.*One-shot mode*: However, if the measurement speed is in the main focus of interest, a “one-shot” MIMO measurement should be applied, where all transmitters are operated in parallel according to (Equation 1). In this case, other types of binary PRN codes with improved cross-correlation properties are additionally required [24], which can be easily achieved by adjusting the feedback structure of the shift registers in stimulus generation, while the remainder of the UWB sensor structure, nevertheless, is not affected by changing the PRN type. The autocorrelation properties of such PRN codes are, however, less perfect as those of an M-sequence. Examples of such sequences, which can be generated from configurable shift registers, are *Gold* or *Kasami* codes. Attention should be paid with respect to receiver saturation. Since all transmitters are working in parallel, the power of the individual transmitters has to be reduced compared to the sequential mode of operation due to the power accumulation at the receiver inputs. This will affect the SNR, so that, finally, the speed advantage of the one-shot mode compared to the sequential mode is often insignificant (see [10] at chapter 3.3.6.2).*Mixed mode*: This mode combines the modes previously provided. While all Rx antennas are still receiving continuously, a partition of Tx antennas may be grouped to a subarray that synchronously transmits the identical M-sequence in parallel and then is repeatedly switched to form a new subarray of transmitting antennas. This combination of parallel and sequential modes may be used to increase the transmitted power that is focused in a certain spatial direction as it is a beamforming array to a certain understanding.

Regarding the parallel operation of an increasing number of Rx channels, the accumulating amount of data quickly becomes a problem as it is difficult to handle huge amounts of data in a realtime processing scenario. This might be elucidated best by the following example focusing on the data amount of one single channel:

Given a scenario where a single measurement channel is operated with the following configuration settings:The stimulus is a 9th-order M-sequence with a length of $N=2^{9}-1=511$ chips.The M_CLK is set to $f_{0}=7\mathrm{GHz}$.By dividing M_CLK through S=512, it concludes $f_{S}\approx 13.7\mathrm{MHz}$.The ADC samples at $f_{S}$ with a resolution of 16 bit.

Due to the implemented subsampling, per DIV_CLK cycle TS=1fS=512·1f0≈73 ns, only one chip of the sent sequence is sampled. Following from that, to reconstruct the whole stimulus sequence once, 511 successive samples are required, thus resulting in ≈37.3 μs recording time $T_{R}$. If this duration is extrapolated to one whole second, approximately 26,800 complete sequences may be recorded and consequently cross-correlated to obtain the impulse responses. This quantity is also understood as the measurement rate mR=1TR of a channel, which is specified in IRF/s. Considering the amount of data per second, one single channel records at the highest possible measurement rate, 13.7MSample/s are acquired which corresponds to a data volume of ≈27.4 MB of raw data per channel and per second.

In many cases, the high measurement rate just found is not required at all. Hence, depending on the time variance of the targeted application, it may be decreased significantly. Since the sampling occurs in a fixed timing grid, the measurement rate and, thus, also the expected amount of data cannot be dropped by reducing the sampling rate. It is, however, possible to reduce the resulting data volume by caching and pre-processing the data, applying one of the various data reduction strategies as, e.g., synchronous averaging in the case of M-sequence radar. Therefore, it is quite self-evident that, at every measurement node, first level data reduction and SNR enhancement should be realized in the nearby vicinity using digital pre-processors. Another conclusion that can be drawn from this, again, is the importance of the system synchronicity, as any principle of synchronous data processing is based on accumulating the identical chips of a sequence in each iteration.

Summarizing the key requirements of a MIMO imaging system built according to the M-sequence measurement principle the following criteria have to be satisfied:Determining the system’s overall jitter performance, there has to be one single fast master clock, which has to be exceptionally stable.Derived from this fast master clock, there is a set of divided clock signals that appears in a fixed grid to the master clock. While the timing of the T&H sampling is critical in terms of the sampling jitter (picosecond precision), the timing of the ADC sampling clock is less critical (nanosecond precision), once there is only a slow decay during the T&H hold phase.Synchronicity is the most significant aspect, not only in terms of the coherent generation of sequences or samples throughout an arbitrary number of channels but also in terms of synchronous pre-processing to reduce the bulk of data and increase the dynamics of the system. Therefore, the timing of an arbitrary number of measurement nodes has to be precisely aligned by a synchronization signal, and each unit requires reset mimics to allow it to return to its defined initial state.

### 2.3. Configurable PRN Generator and Clock Divider Circuits in Integrated SiGe-Technology

One particularly challenging problem is the synchronous generation of the required transmit PRN sequences and clock signals, especially when the system clock speed $f_{0}$ ranges well into the multiple tens of GHz. Within one single clock cycle, all sequence channels and clock signals must align precisely upon the occurrence of an external synchronization event. To this date, only very high-end field-programmable gate array (FPGA) circuits are capable of providing this functionality at the disadvantage of their prohibitively high power consumption, integration effort, and price tag.

In ongoing research into a novel CS-based UWB radar principle [19], the *XG1 (“Xampling Generator”)* integrated device was designed and implemented to solve this problem. The device offers one fully configurable LFSR of arbitrary sequence order up to 24 and a configurable integer-*S* clock divider for arbitrary S=8⋯1048583, which operates at a clock rate of up to $f_{0}=30\mathrm{GHz}$. With a size of only 0.8 mm${}^{2}$ and a power consumption of <1 W, the device outperforms any FPGA-based solution in terms of power and area consumption, as well as economic viability.

Both output signals are generated as strictly synchronous to the provided CLK signal and can be reliably synchronized to the falling edge event of the external SYNC input. While this is mandatory for generating analog control signals for implementing deterministic measurement operators (Φ in the context of [19]), it is also particularly well suited for our proposed synchronous MIMO PRN radar system. The reconfigurable LFSR is well suited to generate orthogonal transmit sequences with good cross-correlation properties, such as *Gold* codes, which is particularly needed in *one-shot* mode. Further, choosing the clock divider ratio arbitrarily offers greater flexibility in better matching $f_{0}$ and $f_{S}$ to the ADC capabilities, as the equivalent-time subsampling principle works with any divider ratio *S*, as long as *S* does not divide *N*.

Figure 5 shows the block diagram and micrograph of the device as well as introducing a timing scheme describing the synchronization process. Internally, the device is capable of operating at very high relative speeds of up to 16-th of the technology node’s transit frequency $f_{t}$. This is only possible through the introduction of an internal secondary clock domain running at $f_{0}\;/\;4$, from which the generators each produce symbols of four consecutive output bits per secondary clock cycle. Specialized architectures for LFSR generation and clock division ensure the reliable generation of the correct symbol streams within the timing constraints of the relaxed secondary clock cycle time [25,26]. A 1:4-serializer then synthesizes the correct output bitstreams from these symbol streams and cleans the output from any data signal jitter that might occur due to the logic paths of different lengths within the symbol stream generators. This ensures superior signal integrity, with wide-open and consistent eye diagram performance, regardless of the chosen configuration. An inter-integrated circuit (I${}^{2}$C) device controller, supporting transparent control signal access and daisy-chained dynamic device addressing, allows for the area-efficient in-system configuration of all XG1 devices in a multichannel application.

The LFSR symbol stream generator is designed in a way that it can easily be extended to ever-increasing register sizes (corresponding to maximum LFSR order) with only a minimal penalty on the register speed per doubling of the register length. This is particularly useful when aiming to generate transmit sequences more sophisticated than mere M-sequences. The time domain product of any set of arbitrary ranging codes $m_{1}\ldots k_{\Omega }$ (corresponding to the *xor*-operation of their digital bitstreams) can be generated from a single LFSR of order $m=\sum_{i=1}^{\Omega }m_{i}$. In the case of the *Gold* codes, which are prominently used for their superior low cross-correlation, one LFSR of order $2m_{g}$ can generate a set of gold codes with period $2^{m_{g}}-1$. As another example, superposing multiple M-sequences of different order results in ranging capabilities of an extreme ambiguity range [27,28]. If combined with the extreme bandwidths available in millimeter-wave bands, this inspires the conception of multichannel ranging systems capable of extreme precision, range, and high spatial resolution. As a side remark, the current reduction efficiency of the partial shut-off scheme employed in the reconfigurable LFSR improves with increasing register size (due to drastically increased degrees of freedom) [25], increasing the significance of power savings in such applications.

## 3. System Architecture

Building upon the generic design aspects of MIMO UWB sensor devices (Section 2.2), in this section, the reader is familiarized with the proposed system approach, which also incorporates the introduced *XG1 “Xampling Generator”* (Section 2.3). Proceeding from the basic sensor module, which is referred to as *“standalone measurement node”*, the reader learns about the underlying clocking and synchronization scheme that can be scaled to control an arbitrary number of channels.

### 3.1. Standalone Measurement Node

Representing the sensor approach in a much more detailed way, Figure 6 provides a block schematic on how the generic system functions of the basic standalone measurement node are put into practice. For transmitting or receiving, respectively, there are four antennas at each measurement node (two are Tx and two are Rx), which are combined in the so-called antenna unit. These antennas can be arbitrarily arrayed so that, e.g., dual-polarimetric (i.e., two bistatic antenna configurations—one vertically polarized and one horizontally polarized) measurements can be implemented.

The antenna front end is followed by the transmit and receive (*“TxRx”*) unit, which deals with the generation of stimulus signals and the reception of back-scattered signals. For these purposes, two custom application-specific integrated circuits (ASICs) are utilized. Each Tx signal is generated by the so-called *XG1 “Xampling Generator”*, which is presented in Section 2.3. Equipped with an I${}^{2}$C interface, these ICs output arbitrarily programmed PRN sequences as well as a settable divided clock signal. Besides from Tx, each Rx signal is received by a custom T&H circuit called *“IS-TH0508”*. This circuit has a large input bandwidth of about 6 GHz in baseband and accepts clock inputs as slow as a few MHz [29] (i.e., there is a slow decay of the signal during the hold phase), which is particularly needed for subsampling. Strongly deviating from the state-of-the-art sensor approach, which is presented in Section 2.1, in this proposed design, the sampling clock controlling the T&H is not identical to the clock controlling the ADC sampling. Instead, the divided clock output FS running at $f_{S}$ of each XG1 Tx chip is applied to control the timing of an associated T&H circuit. That way, not only the T&H sampling has the same appropriate jitter performance as the output PRN sequence, but also the synchronicity of the transmitted sequences and received response signals may be inherently provided.

The rigid timing framework of all clocked system properties originates from a single RF master clock circuit. This circuit, the main part of which is the *“LMX2595”* clock synthesizer, provides a stable M_CLK signal (frequency $f_{0}$) that is adaptive within a huge range of 0.01–20 GHz by applying a frequency divider, voltage-controlled oscillator (VCO) or frequency doubler functions [30]. Its intended output signal parameters are controlled via serial peripheral interface (SPI).

Within a dedicated synchronization and clock distribution unit, the M_CLK signal becomes distributed to each LFSR allocated at the XG1 Tx units. Besides from that, a divided clock (DIV_CLK) at $f_{S}$ is obtained from M_CLK through clock division. This signal is then distributed to control ADC sampling as well as it clocks the circuitry responsible for the generation of the steep-edged synchronization pulse. Note that, although in the system there are several divided clock signals—FS originating from each XG1 Tx chip and DIV_CLK originating from the synchronization unit—it is intended that both are operated at the same frequency $f_{S}$ but with a different stable phase with respect to the parenting master clock at $f_{0}$. The SYNC signal is the third signal being distributed by the synchronization unit. As described earlier, it does on the one hand start the synchronization of multiple XG1 Tx chips, but on the other hand it also triggers the data recording at the utilized acquisition system. In this way, the start of a transmitted sequence is always synchronous with the recording of the received samples, which is also mandatory in particular for various applicable data reduction methods in the pre-processors.

The open-source data acquisition platform *“Red Pitaya^*®*^ SDRlab 122-16”* (briefly referred to as Red Pitaya (RP)) [31] serves as the core unit responsible for analog-to-digital conversion and data pre-processing. It features two 16-bit-wide ADC inputs with a matched impedance of 50 Ω and a maximum sampling rate of $f_{S_{max}}=125\mathrm{MHz}$. In the targeted use case, both the ADC channels are clocked from an external clock source that is provided as DIV_CLK by the synchronization unit and can in addition be delayed by a programmable delay circuit. Since this external clock signal is also redistributed within the RP platform to clock further system components, it is obvious that this signal must be supplied continuously. It follows that the validity of the recorded data must be initiated by a trigger signal so that the invalid data recorded before the trigger can be discarded. All functional properties of the RP are controlled by the *“Zync^*®*^ XC7Z020-1”* system on chip (SoC) that combines a FPGA with a dual-core *“ARM Cortex^*®*^-A9 MPCore”* processing unit. That way, not only hardware–level signal manipulations may be implemented on the FPGA but also various operating systems (e.g., Linux) on the processor unit may be used for the simple control of the wide variety of system functions (e.g., control of output data stream to a secondary processing unit via the Ethernet (ETH) interface, on-chip memory access, etc.). An example of the desired user-defined software is the application of signal processing functions such as impulse compression, which can be implemented on the Zync^®^ processing system with the help of the Python library fastmat [32], which is capable of handling the highly structured algebraic operations behind retrieving H efficiently, using matrix-free methods.

In order to control and modify the overall measurement operation, an additional control unit called *“measurement platform”* is added to the standalone measurement node. Via the universal serial bus (USB) interface, this unit connects to the personal computer (PC) acting as secondary processing or host unit. On that PC by using the graphical user interface (GUI), the user sets the parameters for the envisaged measurement. By means of various digital interfaces, the selected settings are transferred to each of the units. Using the SPI, a predefined dataset is assigned to the LMX2995 master clock generator, which then adapts its output frequency $f_{0}$ to the chosen value or waveform. In addition, the measurement platform is equipped with an I${}^{2}$C interface that handles settings and control mechanisms of each XG1 sequence transmitter chip. That way, every single Tx may be adapted to apply one-shot (parallel orthogonal Tx sequences), sequential (switched single Tx), or mixed (switched Tx subarray) measurement modes and each of the clock dividers is set to meet the wanted track and hold timing pattern. Besides these standardized digital interfaces, the measurement platform accommodates several general purpose input/output
(GPIO) ports, some of which are used especially for controlling the entire measurement procedure. These signals are:TRIG, which starts the synchronization circuitry and thus triggers the generation of synchronous sequences as well as the valid recording of ADC samples;BUSY, which is set to prevent faulty resynchronization while a measurement is still running;RST, which resets the synchronization circuit to its initial state.

These are all transferred to the synchronization unit.

### 3.2. Synchronization and Timing

The problem that arises from these GPIO control signals being CMOS–level signals is that they are classified as “rather slow”, resulting in rise and fall times being in the nanosecond range. Additionally, they behave asynchronously compared to the system clock M_CLK. However, to ensure a stable synchronization procedure at each of the generator ICs, the edge of the synchronization pulse must be so steep that one low-to-high transition (or vice versa) requires less than one single fast M_CLK cycle (e.g., $f_{0}=10\mathrm{GHz}$ leads to a period time of $T_{0}=100\;ps$ of one M_CLK cycle). This ensures, among other things, that the timing constraints of the XG1 latches are not violated and undesirable effects (e.g., glitching, metastability, or mismatch—see Figure 7) of faulty synchronization between different channels are avoided.

The metastability problem should especially be addressed, as it occurs whenever there are crossings of synchronous and asynchronous signal domains. As in [33], the metastability is often explained by employing the figure of the minigolf obstacle where a hill separates two valleys. In analogy to the working principle of flip-flops, the valleys represent the stable states (logic 1 and logic 0). By applying a certain force to the golf ball (signal at the flip-flop input), it will either go to the side across the hill or it will stay on the side it started. However, there is the slightest chance that the ball stops on top of the hill, which in the case of flip-flops represents the undefined in-between state. Given the fact that only a small influence such as wind (or, e.g., thermal noise speaking of flip-flop electronics) is sufficient for the ball to roll down into either one of the stable states, this state only is considered as *“metastable”*. Mainly caused by signal transitions happening in a certain time window $T_{w}$ close to the switching clock edge—bounded by the constraints of *“setup and hold times”*—in real circuits, this effect is noticeable by an increased clock-to-output propagation delay. As this describes the additional time it requires for the output to change after a rising clock edge, it can be concluded that it takes a certain time until the flip-flop can resolve the metastable state and consequently returns to a permitted binary state. Since the metastability resolution time $T_{res}$ depends on several parameters (see, e.g., [34]), the unknown delays at the flip-flop outputs may cause a system of varying clock domains to be vulnerable to unstable synchronization and failure.

To counteract the unwanted timing errors, special care has to be applied when generating a suitable synchronization pulse signal. Below, we provide a detailed insight into how the synchronization circuitry used, circumvents or at least minimizes the likelihood of timing errors.

The synchronization unit (briefly named *“sync. unit”*) builds the core unit of distributing all the needed clock and pulse signals that enable the rigid timing scheme. Additionally, it enables the user to add an arbitrary number of channels to the system, which is achieved by a star-shaped distribution of a certain number of “basic outputs” that can be redistributed subsequently. A multichannel system may not be considered as “true MIMO” until there is strict coherence between the different measurement channels. This coherence is ensured by the all-encompassing synchronization of every unit with the help of the sync. unit.

As implemented, the sync. unit is depicted as a functional block diagram in Figure 8 and its underlying timing chart is presented in Figure 9. Encoded by four different colors, the main functionalities are grouped in the circuit diagram (Figure 8). Starting, red symbolizes the fast M_CLK signal at $f_{0}$, which originates from the master clock generator unit and is redistributed using high-speed current mode logic (CML) clock fanout buffers. Derived from M_CLK, the DIV_CLK signal is obtained from a custom-ASIC fixed-*S* clock divider *“IS-CKDIV0508”*, resulting in a rate of $f_{S}=f_{0}\;/\;512$ [35]. This DIV_CLK signal (encoded in blue) not only becomes distributed as differential low-voltage postive emitter-coupled logic (LVPECL) signals to externally clock the Red Pitaya^®^ ADCs but also this signal is distributed within the synchronization unit to control the timing of the D flip-flops used in the synchronization logic. The synchronization logic (encoded in green) and the reset logic (encoded in violet) are the parts of the circuit responsible for converting the slow CMOS control signals of the measurement platform into differential, steep-edged synchronization control signals that start the XG1 LFSRs with fixed predictable timing or prevent faulty resynchronization. To achieve this, fast logic gates, flip-flops, and fanout buffers are used that support the clock speeds of up to 7 GHz or beyond by means of differential transmission standards (e.g., CML, LVPECL) and thus enable edge rise and fall times $t_{r,f}$ in the range of about 30–50 ps. The way the synchronization logic flip-flops are clocked by the DIV_CLK signal further emphasizes there is a tight relation to the master clock $f_{0}$.

How the synchronization is finally performed may be best explained considering the given signal chart (Figure 9). Before a trigger signal is set, the flip-flops should always be reset first to return them to their initial state. This is accomplished by setting the RST GPIO signal of the measurement platform to high state. At the sync. unit, this single ended signal is then level-shifted to a faster differential LVPECL signal F_RST, which is consecutively distributed to the asynchronous reset input of each of the D flip-flops. Once the RST signal is released, the flip-flops asynchronously drop to their output zero state and the reset procedure is completed.

Subsequently, the trigger signal may be set by switching the TRIG GPIO to its high state. The TRIG signal has, again, to be level-shifted first, resulting in the mid-speed trigger signal F_TRIG. What follows is a so-called *“two–flop–synchronizer”* (*DFF1*, *DFF2*), which can be considered the least-complex state-of-the-art synchronizer circuit [33,36]. The goal of the synchronization is to ensure that any metastability that may occur is resolved within a desired time interval $T_{res}$. Since, in the present case, we want to achieve both strict the coherence of an arbitrary number of Tx sequences and, for each measurement node, a stable triggered recording of samples, there are two levels of synchronization. Hence, as there are two levels of signal precision (M_CLK—picosecond precision; DIV_CLK—nanosecond precision), there are also two levels of metastability resolution time. The sync. unit must therefore be designed in such a way that, in addition to the criteria of the nanosecond metastability resolution (nanosecond level of synchronization), it can partially satisfy the criteria of the faster picosecond metastability resolution (picosecond level of synchronization). On the one hand, the synchronicity has only to be as precise as one DIV_CLK cycle leading to the requirement that TresS=1fS=TS is sufficient, but, on the other hand, considering the XG1 synchronization procedure, the resulting SYNC output signal transitions have to be sufficiently steep ($t_{r,f}<T_{0}$), so that, in the faster $f_{0}$ range, Tres0=1f0=T0 can be kept at the synchronizer of each XG1 ic. What happens at the sync. unit’s two–flop–synchronizer is that, even if the first flip-flop *DFF1* became metastable, this metastability would resolve within one DIV_CLK cycle so that the output Q_FF2 of *DFF2* almost never becomes metastable (i.e., the possibility of failure is extremely low or the mean time between failures (MTBF) is extraordinarily large). Resulting from this synchronization step, the now $f_{S}$- as well as $f_{0}$-synchronous edge present at output Q_FF2 is then further distributed using a fanout buffer. While one exit directly connects to a high speed *and*-gate, the second exit is negated and then connected to a third flip-flop (*DFF3*), the output Q_FF3 of which also connects to the logic *and*-gate. The resulting signal at the output of the logic *and* is one single low-to-high-to-low SYNC pulse, which, on the one hand, is perfectly aligned to one DIV_CLK cycle or the duration of 512 M_CLK cycles $T_{\mathrm{SYNC}}=T_{S}=512\;\cdot \;T_{0}$ and, on the other hand, exhibits edge rise and fall times in the required range ($t_{r,f}<T_{0}$). This steep-edged pulse afterwards is redistributed to each XG1 transmitter starting the internal latching (picosecond level of synchronization, see Section 2.3) as well as it triggers data recording at each of the RP acquisition platforms.

After successful synchronization, the BUSY signal may be set to prevent a faulty resynchronization. This is achieved the same way as RST by asynchronously deactivating each flip-flop’s output capability.

To summarize the overall system timing, Figure 10 provides an overview of clock domains, synchronization signals, and the resulting timing order of sampling. Note that, although all signals are related to M_CLK, this signal is not shown at its actual scale.

Once the SYNC signal is received at the XG1 Tx chips as well as the RP acquisition platforms, a measurement is initiated. Apart from the XG1, where the falling edge event of the SYNC pulse starts the synchronized output of the PRN sequence SEQ and divided clock FS, the validated data recording at each of the RPs is triggered (ACQ_ST) by the rising edge event of the SYNC signal.

The third section of the signal chart (Figure 10) depicts the underlying hierarchy of the sampling signals. The first level of sampling is implemented with the T&H circuits of high input bandwidth. Dependent on $f_{0}$ and started by SYNC, each T&H is clocked by the FS signal of an associated XG1 transmitter chip at frequency $f_{S}$. Afterwards, the second level of sampling is realized with the ADCs of the RP platform. These are externally clocked by the DIV_CLK signal of the sync. unit, running at frequency $f_{S}$. However, in order to achieve the optimum sampling time for each selected system M_CLK frequency $f_{0}$, it is necessary that the ADC clock of each measurement node can be shifted within a certain range. Therefore, each of the RP units is enhanced by an adaptive delay circuit. The optimum sampling time may be understood as the time at which the transient of the T&H switching has already abated and a stable but slowly decaying hold value has been established. According to [29], for the given M_CLK frequency $f_{0}=7\mathrm{GHz}$, the optimum sampling time for an ADC connected downstream of the *IS-TH0508* is obtained by delaying the ADC clock ≈4 ns with respect to the T&H sampling clock.

### 3.3. MIMO Enhancement

Concluding from the system’s synchronization architecture, there is the possibility to add further measurement nodes to the system that need the three essential control and clock signals, M_CLK, DIV_CLK, and SYNC, to be properly operated.

Determined by the number of basic outputs, the realized sync. unit is yet designed to operate two measurement nodes in parallel, which results in four parallel channels, each consisting of a Tx and associated Rx.

As seen in Figure 11, these basic outputs are distributed in a star-wise manner. A desired extension of the imaging system by additional nodes leads to the requirement that the additionally required clock and synchronization signals may be obtained using a tree-structured distribution. However, it should be noted that the overall wire lengths (i.e., wire propagation delays) of the clock and synchronization signal lines must be kept the same for each measurement node to be operated in a strictly synchronous manner.

Apart from the time-critical clock and synchronization lines, the distribution of control signals (e.g., the I${}^{2}$C control interface bus that connects to each TxRx unit) is less critical, as these are defined bus systems where adding further elements is always possible within the scope of the bus specifications. Additionally, to ensure that each of the reduced data streams of each measurement node can be further processed, the Ethernet connection to the host unit or PC must be established by using suitable network switch hardware.

## 4. System Performance Evaluation

In order to verify the system conception, an experimental test setup based on two transmitters and two receivers comprising one measurement node was implemented. Figure 6 shows the related block schematics and Figure 12 depicts its practical implementation. The major components are:The sensor node—compare to Figure 4 and Figure 6—containing the data capturing and pre-processing unit (*Red Pitaya^®^ SDRlab 122-16*) as well as the custom-designed SiGe circuits for signal generation (*XG1 “Xampling Generator”*) and RF sampling (*IS-TH0508*);The synchronization unit (Figure 8);The control unit (*measurement platform*);The master clock synthesizer (*LMX2595*);A laptop PC as host unit for data acquisition, visualization, and control.

The aim of the tests is to demonstrate the applicability of the concept for wideband radar applications as well as its timing performance underlining its suitability for UWB array application and weak motion detection. Therefore, each test was conducted in some sense as a back-to-back (B2B) measurement. That is, in order to exclude properties and influences of the transmission path onto the performance evaluation, the Tx and Rx channel are connected directly by RF cables including a 21 dB attenuator to ensure linear operation of the T&H circuit.

### 4.1. Basic Radar Mode of Operation

The aim of the UWB radar device is to determine the impulse response of the transmission path under test. Hence, in the case of a (nearly) ideal transmission path as an RF line, one would expect a δ-pulse-like response of the transmission channel. This is demonstrated in what follows. For demonstration purposes, the master clock rate to drive the LFSR was selected to $f_{0}=7\mathrm{GHz}$, which, finally, also represents the equivalent sampling rate of the receiver channel. The SEQ output of the XG1 is configured to generate a 9th-order M-sequence with a characteristic polynomial $x^{9}+x^{5}+1$, leading to N=511 data samples per impulse response. The duration of the response function is consequently T=Nf0=73 ns and the ambiguity range of the radar becomes about 11 m. The differential output level of the LFSR is about ±700 mV, which directly stimulates the transmission channel under test.

For the sake of simple illustration, a subsampling factor of S=512 was selected for data capturing. This leads to sequential sampling so that the signal samples are captured in their natural order. Note that other subsampling factors provide interleaved sampling, which requires reordering of the captured values. Thus, the actual sampling rate is of about $f_{S}\approx 13.7\mathrm{MHz}$. The recording time for one impulse response requires TR=NfS≈37.3 μs, allowing to record 26,800 impulse responses per second in maximum. The maximum bandwidth of the radar system theoretically available spreads over $B=f_{0}\;/\;N\ldots f_{0}\;/\;2\approx 13.7\ldots 3500\mathrm{MHz}$.

Figure 13 depicts the results for one arbitrarily selected transmission channel as well as the applied measurement configuration. The upper plot (Figure 13b) shows the captured signal. It is actually a pseudo-random signal. However, within longer signal sections with constant amplitude, one can observe an exponentially decaying voltage. This is due to the AC coupling of the T&H circuit and ADC. The T&H input is protected by a DC block of 100 MHz cut-off [29] and the ADC input is operated with a Balun of about 300kHz cut-off frequency [31]. Both measures cut the lowest 10 spectral lines of the signal spectrum––the DC block in the RF domain and the Balun in the sampling domain. In practical radar applications, this band limitation is of no concern, since the antennas typically do not work at such low frequencies. The voltage values provided in Figure 13b refer to the ADC input. Respecting the internal gain of the T&H circuit of 15 dB and the inserted attenuators, the actual RF voltage being present at T&H inputs is in the range of ±20 mV.

Figure 13c represents the captured signal after impulse compression based on the correlation with the ideal M-sequence (which is known from the LFSR polynomial value set at the XG1 IC). The result is a δ-like function with an undershoot due to the AC coupling. The width of the pulse is τFWHM=1f0≈140 ps (full wave half max (FWHM)). The pulse base is nicely flat without any peaks. This indicates that the analog transmission channel including LFSR, T&H, and ADC is not affected by nonlinear disturbances [10]. The time position of the δ-pulse is arbitrarily selected. It can be placed anywhere by an appropriate choice of the starting value of the LFSR or the ideal M-sequence respectively.

In order to evaluate the noise performance, the measurement was repeated several times, leading to an ensemble of 32 response functions $C_{xy_{i}}\left(\tau \right);\;i=1\ldots 32$ as depicted in Figure 13c. Their standard deviation shows Figure 13d. We can observe a constant noise contribution along the whole signal. The remaining randomness of the horizontal line is due to the limited size of the measurement ensemble. It is interesting to note that there is no elevation of noise at the position of the δ-pulse as usual, e.g., in the case of standard pulse systems [6]. The resulting SNR is provided by the expected value of the pulse peak and the average value of the baseline variance: (8)$${\mathrm{SNR}}_{0}=10\log_{10}\left(\frac{E^{2}\left\{\max(C_{xy_{i}}\left(\tau \right))\right\}}{\bar{\mathrm{var}\left\{C_{xy_{i}}\left(\tau \right)\right\}}}\right)\approx 67\mathrm{dB}.$$

Concerning the selected operational parameters of the radar system, the recording time of one response function is about 37 μs (see above). This would allow for measuring micro-motions of up to about 13 kHz oscillation frequency or targets with a maximum speed of up to fourfold the speed of sound [10]. Most of the practical applications do not require such high measurement rates due to the measurement scenario being less time-variant. Therefore, the recording time may be extended in order to repeat the measurements and to subsequently average the data synchronously so that the additional noise is gradually being suppressed. The resulting SNR value is provided by: (9)$$\mathrm{SNR}={\mathrm{SNR}}_{0}+10\log_{10}\left(\frac{T_{M}}{T_{R}}\right),$$
where $T_{M}$ represents the actual time to capture one response function including all averages.

### 4.2. Jitter Performance

Random jitter mainly affects the performance to detect weak target motions or other variations of target reflectivity as well as the suppression of strong static targets by background subtraction (see (Equation 5)). Assuming a stable clock rate $f_{0}$ of the radar system, the randomness of trigger events within the LFSR and of the T&H circuit are to be seen as the major jitter sources [37].

In contrast to additive random noise, jitter only affects signal edges. This typically leads to time-variant noise of the receive signal, which will be spread over the whole compressed radar signal $C_{xy}\left(\tau \right)$ due to the impulse compression [6]. That is the reason why there is not an elevation of noise around impulse peaks (refer also to Figure 13d).

To evaluate the jitter, we therefore had a deeper look into the received data by analyzing the time-dependent noise, which composes from the time-independent additive noise and the time-dependent noise generated by the jitter. The strength of the latter is determined by the strength of the jitter and the slew rate of the signals. Hence, to visualize the resulting effect, one needs a sufficiently large number of data samples on the rising or falling signal edges. Unfortunately, in favor of generating low data volume and simple system complexity, the radar concept uses the minimum possible sampling point density without violation of the Nyquist sampling theorem. Therefore, data points only sparsely populate the signal edges.

To increase the density of data points without any change in the system conception, an adjustable mechanical delay line was inserted into one of the two possible transmission paths—see Figure 14. A series of 86 experiments with successively increasing delays was then performed, each comprising an ensemble of 32 measurements for statistical analysis. The step size of the additional delay was selected to $t_{d}=4\mathrm{ps}$, leading to a delay variation of 0–344 ps spanning all 86 experiments. This covers more than two periods of the RF clock rate. By merging and restructuring the results of all 86 experiments to a single plot—see Figure 14b—one obtains an ensemble of 32 signals being oversampled at an equivalent sampling rate of $f_{os}=250\mathrm{GHz}$, whereat the sampling interval is not strictly equidistant. The expected value and standard deviation of a short signal section are summarized in Figure 14b,d and the zoomed data of one single edge is presented in Figure 14c,e. For illustration, the magenta-colored asterisks emphasize the actually captured data samples under normal operational conditions. The curve of the standard deviation is again corrupted by some randomness, which is due to the small size of the measurement ensemble. However, a clear elevation of the noise level is not observable at the locations of the signal edges. This indicates the dominance of additive noise in favor of random jitter so that the jitter does not degrade the device performance. The related SNR of the actually received signal (SNRraw=20log10(|Vsigpeak||Vnoiseavg|)) is about 40 dB, which still leaves some space for future circuit improvements. This value is in good accordance with Equation (Equation 8), which additionally includes the correlation gain of $10\log_{10}\left(N\right)\approx 27\mathrm{dB}$ for a 9th-order M-sequence.

### 4.3. Repetitive Array Synchronization Accuracy

As mentioned in Section 2.2, the radar device allows for different array operation modes. In the case of orthogonal transmission signals, the whole radar array, i.e., all transmitters and receivers, must only be started once. The array then captures the data continuously. Another option is to deal with only one stimulus signal (ideally an M-sequence) that is either transmitted by one single transmitter or a beamforming subarray of transmitters. In that case, the transmitters must be operated individually and consecutively. Hence, every transition to a new active transmitter (or to a new subarray) needs a new Rx–Tx–synchronization.

The challenge of the synchronization in both cases—orthogonal or nonorthogonal Tx signals—is a reproducible start of all involved Tx and Rx channels with the precision of the master clock edges at $f_{0}$, while the edge of the trigger signal TRIG (see Figure 8 and Figure 9) releasing the whole measurement cycle has a long rise time, which extends over tens of M_CLK cycles.

For the performance characterization of trigger stability and reproducibility, a series of 1000 measurements was performed, whereat each measurement was started by a new trigger event. Two types of measurements were involved. In the first case (introduced as B2B), only receivers and transmitters within a pair of associated Tx and Rx channels (i.e., they are directly related in terms of the FS signal) were connected. The second test setup (called crosswise back-to-back (XB2B)), in contrary, dealt with the cross-connection between Rx and Tx channels that are not corresponding in terms of the FS clock signal. From all measurements, the impulse compression was performed and the location of the pulse maximum was registered. Figure 15 summarizes the results. It shows histograms of the time position as well as of the magnitude of the largest sample of $C_{xy}\left(\tau \right)$. The upper histograms refer to the time position. Note the logarithmically scaled probability axis. None of the 1000 measurements provoked a deviation of the time position. Hence, all trigger events met perfectly the same M_CLK edge. The histograms in the lower row show the magnitude distribution of the maximum value. As expected, it provides a Gaussian distribution whose width corresponds to the standard deviation as depicted in Figure 13d. The mean values of the different transmission paths are slightly different due to weak variations of T&H gain and LFSR output voltage of the different channels.

## 5. Conclusions

In many fields, there are measurement scenarios where optically opaque SUTs can be observed by different methods of microwave imaging. Hereby, often arbitrarily shaped antenna arrays are employed that, while minimizing the scanning effort, further enhance the imaging in terms of target classification at the same time. This is because of the angular resolution, and different polarizations, features, or objects can also be spatially detected or temporally tracked. However, multichannel systems usually entail an increased circuitry effort, which is particularly noticeable in the synchronization within multiple units and in the processing of the exaggerated amount of data. In addition, it has been shown that, depending on the application, a certain flexibility in the adjustment of the sensor is necessary in order to adapt it to the targeted application.

The system concept proposed in the article enhances the state-of-the-art UWB PRN sensor as it addresses both of these aspects by ensuring tight synchronization based on the novel XG1 LFSR and clock divider chip and by leaving many system parameters adjustable by the user over a wide range.

The core of the system synchronicity is that each clock or pulse signal being distributed within the system originates from a single monotone RF clock. With the help of a synchronization pulse signal that is synchronously generated to that fast RF clock, the synchronous data recording at an arbitrary number of measurement nodes, each consisting of two transmitters and two receivers, can be started. Thereby, the synchronicity is inherently provided within each channel consisting of a Tx and an associated Rx, since each XG1 Tx chip simultaneously provides the synchronous sampling clock for an associated T&H of the receiving path. However, the synchronicity within an arbitrary number of measurement channels is also realized as all of the channels depend on the same fast RF clock as well as on the synchronization pulse signal that starts these units.

Besides the applied synchronization mechanism that allows for multichannel scalability, the proposed system was designed with respect to user-configurability. Hence, adjustable hardware components such as the configurable RF clock synthesizer are applied. Particularly important in terms of adjustability is also the XG1 chip, which allows the user to select any PRN sequence up to the 24th-order and clock dividers up to 104,858. However, not only the hardware but also the software can be adapted to the needs of the application, as the data pre-processing of each measurement node is based on the open-source Red Pitaya^®^ platform.

First, performance tests were carried out with a system prototype comprising two channels (one measurement node). So far, good system dynamics as well as good jitter behavior were observed. Nevertheless, the achieved stability of the synchronization is outstanding, which worked 100% sample-accurate in the conducted continuity tests.

Building on the promising interim results, however, it is still necessary to shed some light on other aspects to ultimately build a multichannel system with many elements. This is particularly aimed at the software and firmware development required to automatically reduce the amount of data in the pre-processors of each of the measurement nodes by means of averaging. This way, the data transfer of each node to the secondary processing or host unit is prevented from being the bottleneck of the overall application. Furthermore, a large number of channels requires an elaborately growing control mimic, which is needed to monitor and control the measurement procedure. Finally, the large amount of data to be acquired requires appropriate signal processing algorithms to decode the information contained, which leads to the application-specific analysis result of the test scenario. These and other developments will be examined in an article based on this one.

## Figures and Tables

**Figure 1 sensors-23-02454-f001:**
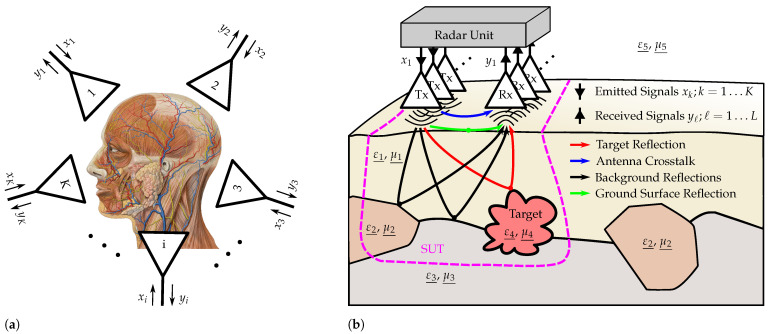
Different applications of MIMO imaging using arbitrarily arranged antenna arrays. (**a**) Medical imaging—conformal array: Detection of head arterial movements (head adapted from [9]); (**b**) MIMO GPR—planar array: Detection of shallow-buried improvised explosive devices (IEDs).

**Figure 2 sensors-23-02454-f002:**
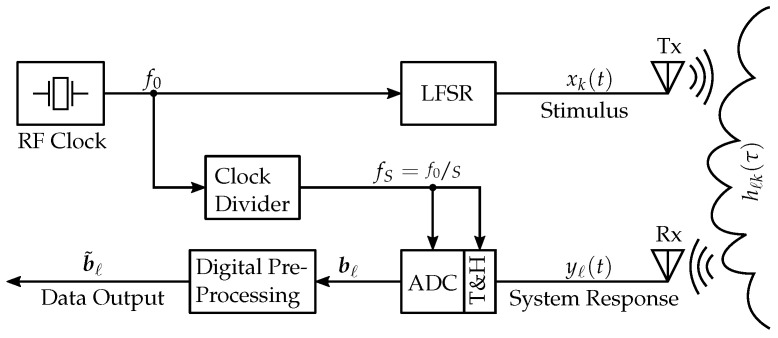
State-of-the-art MLBS UWB sensor block design.

**Figure 3 sensors-23-02454-f003:**
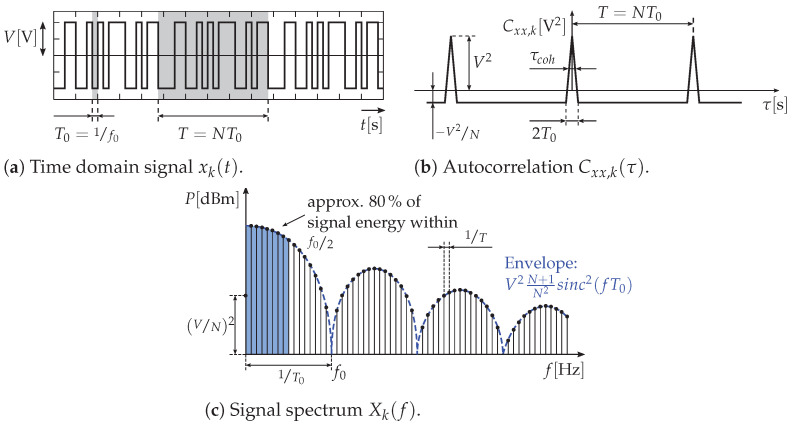
M-sequence signal representation without band limitation.

**Figure 4 sensors-23-02454-f004:**
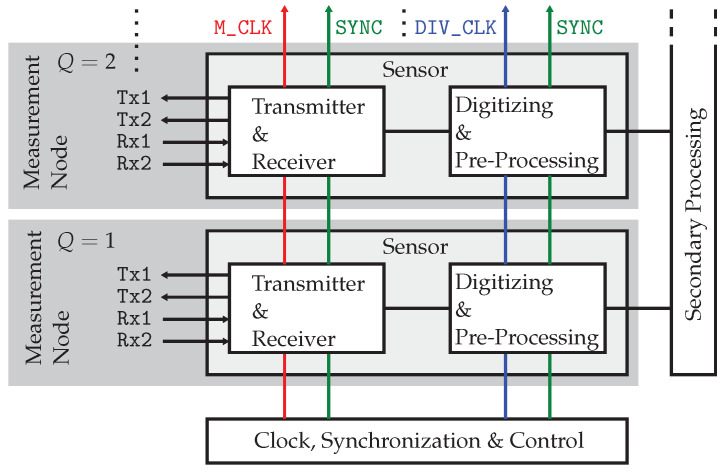
Distribution of clock and synchronization signals in a generic multinode imaging sensor system.

**Figure 5 sensors-23-02454-f005:**
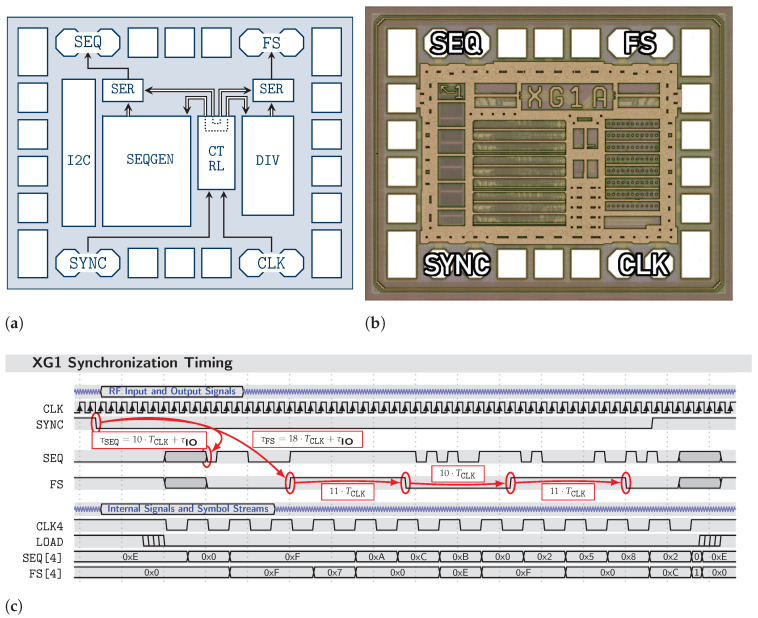
The *XG1* device in 0.13 μm SiGe Technology, with synchronous and fully configurable PRN sequence generator and one integer-S clock divider, designed for operation with $f_{0}$ of up to 30 GHz. (**a**) Block diagram. (**b**) Die-shot with a total area of 0.80 mm${}^{2}$. (**c**) Signal chart for S=f0fS=21, $p\left(x\right)=x^{12}+x^{11}+x^{7}+x^{4}+1$ and starting vector 0xFFF0E, showing the synchronization to an external clock signal CLK and synchronization event SYNC.

**Figure 6 sensors-23-02454-f006:**
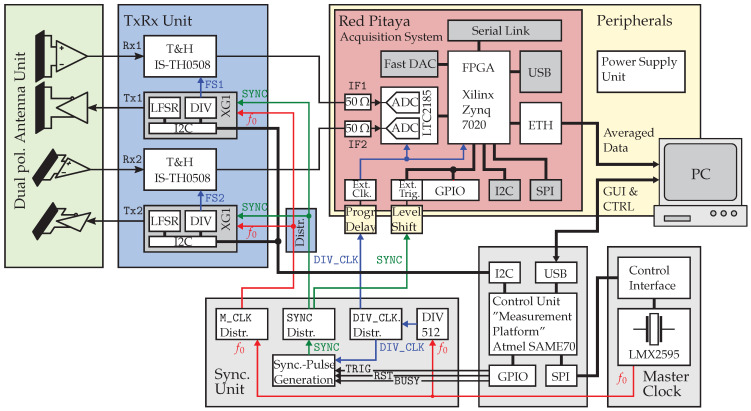
System architecture of the basic standalone imaging node. The colored arrows represent the different signal classes. Red symbolizes the fast M_CLK at frequency $f_{0}$, blue symbolizes the divided clock signals at frequency $f_{S}$, and green marks the signal lines for the synchronization of the units.

**Figure 7 sensors-23-02454-f007:**
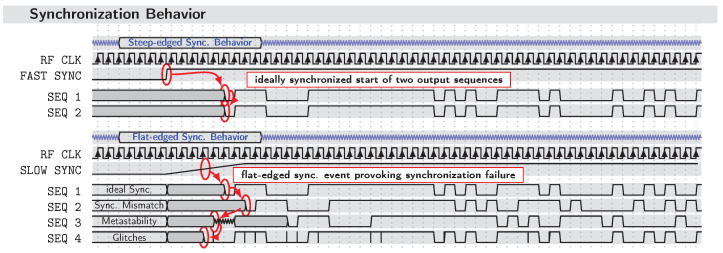
The effect of edge steepness on synchronization behavior within multiple channels. Above: ideal case, i.e., steep synchronization edge at a sufficient distance from the rising clock edge; below: fault-provoking case, i.e., slow synchronization edge covering multiple clock periods.

**Figure 8 sensors-23-02454-f008:**
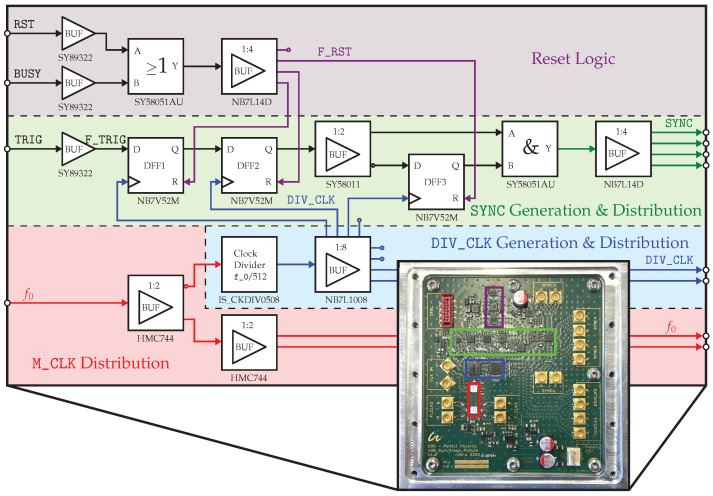
Circuit block design of the synchronization unit. Circuit functions are grouped by color.

**Figure 9 sensors-23-02454-f009:**
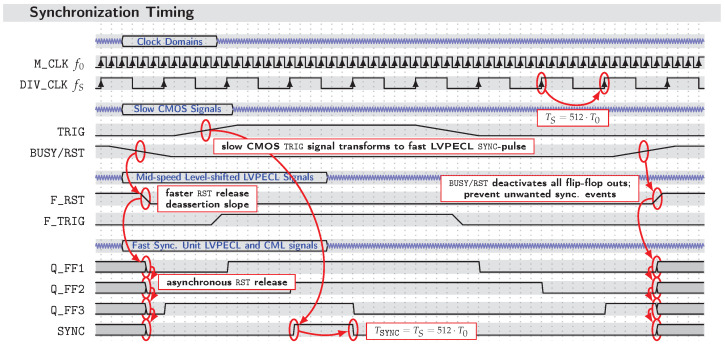
Signal chart for sychronization pulse generation circuit. Note that due to illustration reasons, M_CLK is not shown at the correct scale. One cycle of DIV_CLK actually covers 512 cycles of M_CLK ($T_{S}=512\;\cdot \;T_{0}$).

**Figure 10 sensors-23-02454-f010:**
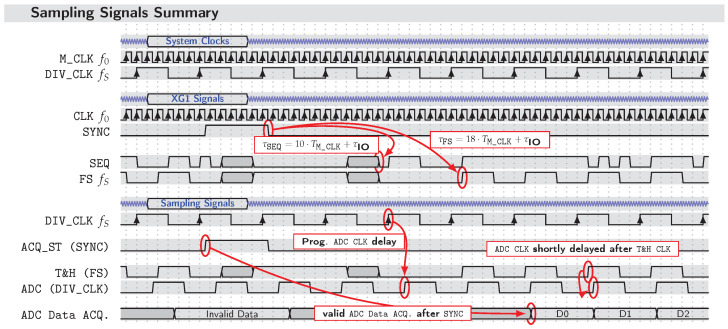
Signal chart summarizing system clock domains, synchronization, and timing of different layers of sampling. Note that, due to illustration reasons, M_CLK is not shown at the correct scale. One cycle of DIV_CLK actually covers 512 cycles of M_CLK ($T_{S}=512\;\cdot \;T_{0}$).

**Figure 11 sensors-23-02454-f011:**
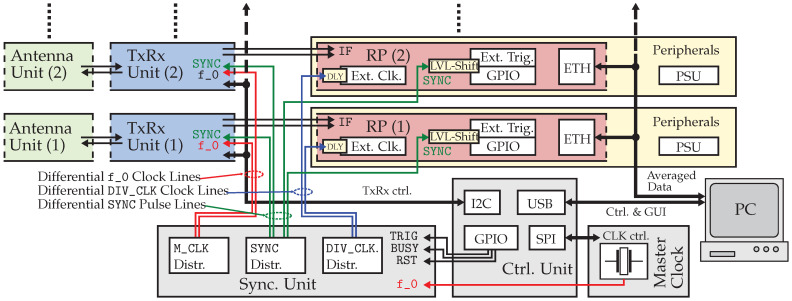
System architecture showing MIMO extension through distribution of clock and synchronization signals.

**Figure 12 sensors-23-02454-f012:**
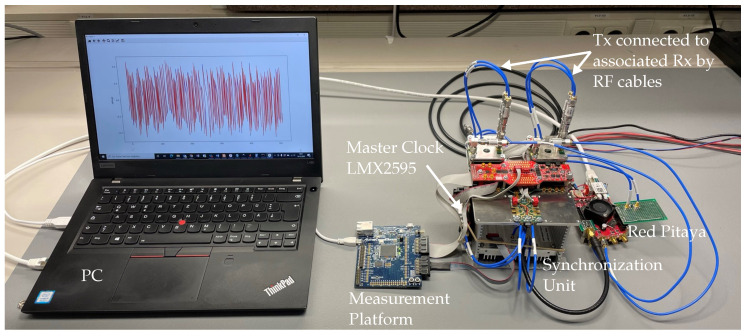
Prototype test setup of the standalone measurement node. Each Tx is back-to-back to its associated Rx with 21 dB signal attenuation. Synchronously recorded M-sequences of 9th order are shown on the screen.

**Figure 13 sensors-23-02454-f013:**
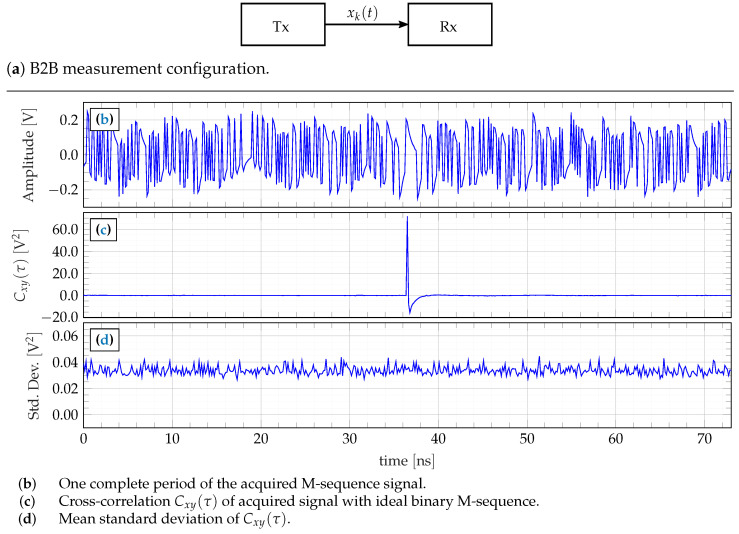
Signal peculiarities of the M-sequence signal measured in back-to-back (B2B) configuration: 9th-order M-sequence driven at $f_{0}=7\mathrm{GHz}$.

**Figure 14 sensors-23-02454-f014:**
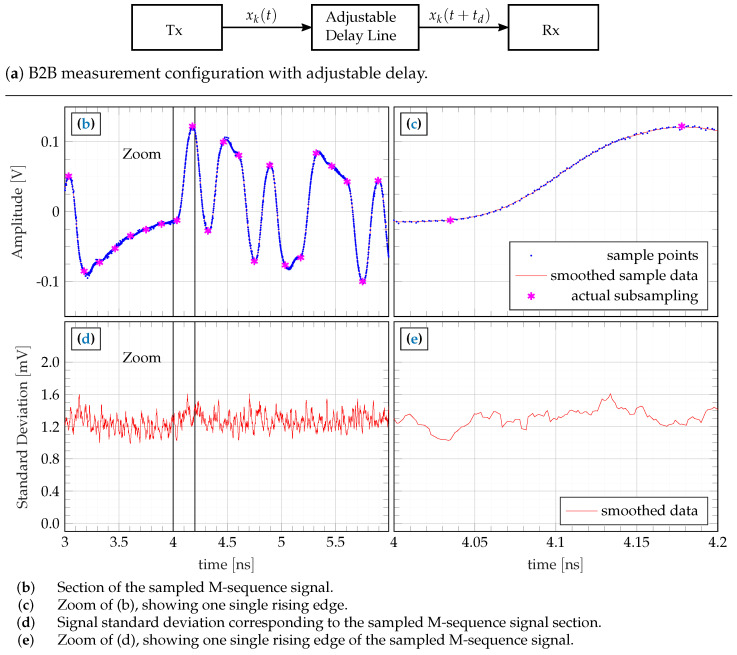
Jitter behavior of the M-sequence signal determined using stepwise incremental retardation (4 ps delay steps) in back-to-back (B2B) operation: 9th-order M-sequence driven at $f_{0}=7\mathrm{GHz}$, resulting oversampling at an equivalent sampling rate of $f_{os}=250\mathrm{GHz}$.

**Figure 15 sensors-23-02454-f015:**
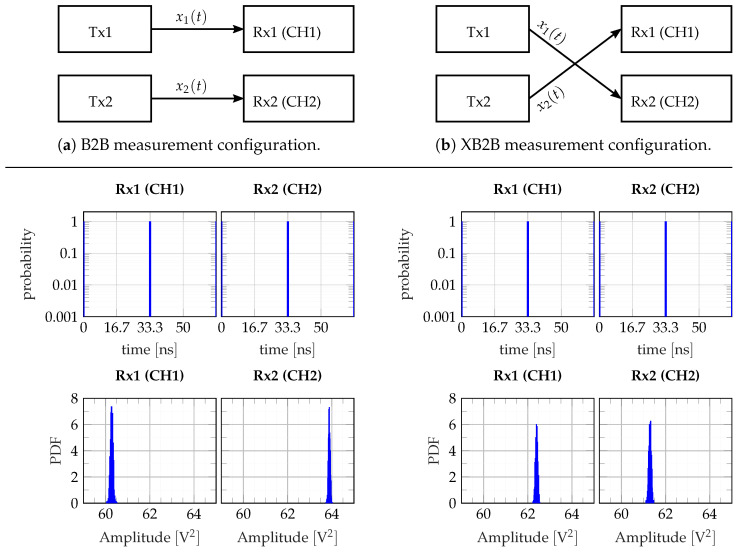
Synchronization stability test using iteratively synchronized data recording in back-to-back (B2B) and crosswise back-to-back (XB2B) configurations: 9th-order M-sequence driven at $f_{0}=7.68\mathrm{GHz}$, 1000 iterations. The diagrams signify the indicated statistical properties of the corresponding pulse peak in terms of pulse peak position in time domain as well as the the distribution of pulse amplitude noise.

## Data Availability

Data sharing is not applicable.

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
