# Peer review of "Configurable Pseudo Noise Radar Imaging System Enabling Synchronous MIMO Channel Extension"

_sensors, 2023, doi:10.3390/s23052454_

Round 1

Reviewer 1 Report

This paper presented a fully synchronized multichannel UWB radar imaging system design approach based on pseudorandom noise sequences. The radar system is provided with adaptive hardware and feasible signal processing open-source framework. This manuscript is well-written and the details of the system design are described comprehensively.

There are several minor comments as follows.

1) In the Introduction, please add the references about the existed approaches of PNS radar system design.

2) In the experiments, only the ideal evaluation is performed when the TX is directly connected to the associated RX by the RF cables. For more robust evaluation, if it is convenient, please add an experiment where the antennas are connected to the TX/RX and a real target is illuminated.

3) In Figure 14, the related SNR is calculated as about 40dB. Could you please add the detailed calculation method for SNR.

4) In Figure 13(c), the time delay of peak value is about 36ns. In this case, the corresponding propagation range is about 1m. Please explain the length of RF cables in Figure 12 to match the propagation range.

Author Response

Please find the point-by-point answer to your comments in the attached rebuttal letter.

Reviewer 2 Report

The paper describes the design of a MIMO UWB radar based on pseudorandom noise (PRN) sequences. The paper is well-written and focused in the synchronization of the channels.

However, I have some comments to improve the work:

1.       Some minor mistakes should be corrected: In (3), hlk(t) should be a function of tau delay parameter,  hkt(tau)

2.       The basic radar operation is shown with the TX and Rx Connected by a coaxial wire with an attenuator. Some measurements with real antennas can be included to show better the radar operation and show the effect of distortion of the antennas.

3.       The radar uses an M-sequence. It is not clear that the spectrum falls within the UWB mask between 3.1-10.6 GHz. A simulation or measurement of the spectrum can be included.

Author Response

(The authors gave the same response as above.)

Reviewer 3 Report

I have no substantial objections to the research, which is well described in the submitted paper. I consider the topic of the submitted paper to be current. The research strategy and results are clearly described. I appreciate the design of the UWB radar architecture and the measurements performed. I consider the derived algorithms for processing UWB radar measurement signals to be correct. I have no fundamental comments on the approximations adopted in their derivation. The paper is considered as a good introduction to the design of an ultra-wideband (UWB) radar system based on pseudo-random noise (PRN) sequences, the key features of which are its user adaptability to meet the requirements required by microwave imaging applications and its multi-channel scalability. The references are appropriate. I recommend publishing this paper after slight modifications.

Author Response

(The authors gave the same response as above.)

Round 2

Reviewer 2 Report

The authors have clarified my comments and inserted some modifications to the manuscript. I have no further comments.